# Next Challenges for the Comprehensive Molecular Characterization of Complex Organic Mixtures in the Field of Sustainable Energy

**DOI:** 10.3390/molecules27248889

**Published:** 2022-12-14

**Authors:** Anthony Abou-Dib, Frédéric Aubriet, Jasmine Hertzog, Lionel Vernex-Loset, Sébastien Schramm, Vincent Carré

**Affiliations:** LCP A2MC, Université de Lorraine, F-57000 Metz, France

**Keywords:** bio-oil, high-resolution mass spectrometry, ion mobility, tandem mass spectrometry, fractionation, derivatization

## Abstract

The conversion of lignocellulosic biomass by pyrolysis or hydrothermal liquefaction gives access to a wide variety of molecules that can be used as fuel or as building blocks in the chemical industry. For such purposes, it is necessary to obtain their detailed chemical composition to adapt the conversion process, including the upgrading steps. Petroleomics has emerged as an integral approach to cover a missing link in the investigation bio-oils and linked products. It relies on ultra-high-resolution mass spectrometry to attempt to unravel the contribution of many compounds in complex samples by a non-targeted approach. The most recent developments in petroleomics partially alter the discriminating nature of the non-targeted analyses. However, a peak referring to one chemical formula possibly hides a forest of isomeric compounds, which may present a large chemical diversity concerning the nature of the chemical functions. This identification of chemical functions is essential in the context of the upgrading of bio-oils. The latest developments dedicated to this analytical challenge will be reviewed and discussed, particularly by integrating ion source features and incorporating new steps in the analytical workflow. The representativeness of the data obtained by the petroleomic approach is still an important issue.

## 1. Introduction

The majority of the worldwide energy output and chemicals derives from petroleum [1]. This source is considered as a very complex organic matrix with hundreds of thousands of compounds and whose composition varies with the location, the extraction and the refining process [2,3,4]. However, it is also responsible for economic, political, and ecological issues, and more sustainable alternatives are currently considered and developed. Among them, the first-generation bio-fuel is the most commercially used. Its production involves the use of human and animal nutrition. Therefore, the second and third bio-fuels were developed and they are produced from lignocellulosic biomass and algae, respectively [5,6,7]. Thus, bio-oils may be produced by extraction and conversion of the lipids from algae, by thermal activation (pyrolysis) or hydrothermal liquefaction (HTL) of lignocellulosic biomass and algae [8]. These materials are very complex organic matrices. Independently of the fuel origin, crude oil or biomass, all these matrices require different upgrading treatments in order to be suitable for engines, with the appropriate physicochemical properties. Indeed, as shown in Table 1, second (2G) and third (3G) generation bio-oils obtained from the thermochemical conversion of biomass present higher oxygen amount than crude oils, which is notably responsible for lower energetic density given by the higher heating value [9,10,11,12].

Thus, bio-oils must undergo deoxygenation and, depending on the feedstock, denitrogenation treatments, to reach physicochemical properties similar to those of petroleum-based oils [7,13]. To assess and optimize the treatment efficiency, the finest composition description of the sample before and after treatment is needed. One of the preferred ways of characterizing the composition is high resolution mass spectrometry (HRMS), which, in application to the non-targeted analysis of petroleum, is called petroleomics. HRMS instruments comprise Q-TOF, Orbitrap, and Fourier transform ion cyclotron resonance mass spectrometer (FT-ICR MS), whose capabilities are later described. Both Orbitrap and FT-ICR analyzers are Fourier Transform Mass Spectrometry (FTMS) instruments [14]. Alan Marshall is a pioneer in petroleum analysis by HRMS and, more specifically, by Fourier transform ion cyclotron resonance mass spectrometry (FT-ICR MS). In association with Ryan Rodgers, he coined the term “petroleomics” to determine petroleum properties and behavior based on description of all detected petroleum components at the molecular level [15,16]. Nowadays, this non-targeted approach is also applied to bio-oils [17].

Due to the high molecular complexity of crude oil and bio-oil samples, HRMS is required with or without using an upstream separative method, such as chromatography. HRMS techniques are Fourier transform ion cyclotron resonance (FT-ICR), Orbitrap, and Time of Flight (TOF) mass spectrometry, whose working principles are well known and reported elsewhere [18]. Petroleomic studies and principles regarding crude oil and bio-oils were reported in different recently-published reviews [19,20,21,22,23,24,25,26]. The high mass resolution and accuracy (Figure 1A,B) allow the technique, respectively, to detect and assign hundreds of thousands of molecular formulae. As an example, atmospheric pressure photo-ionization (APPI) FT-ICR MS analysis of a volcanic asphalt sample demonstrated the ability to assign more than 126,000 chemical formulae [27]. The great amount of generated data requires a graphical representation to facilitate both the sample description and its comparison with another investigated sample. As illustrated in Figure 1C, the molecular assignments can be represented according to the heteroatom class (CH, CHO, CHON…), the double bound equivalent number (DBE) (i.e., the number of unsaturation, vs. the carbon number), and also on the van Krevelen diagram. This latter graph was historically used for coal, petroleum, and kerogen samples but it is now applied to a large range of matrices [28,29]. For instance, the H/C vs. O/C van Krevelen diagram allows the distinguishment of chemical families such as lipids, unsaturated hydrocarbons, phenolics, and carbohydrates [28]. This diagram represents the different compounds by dots whose x and y coordinates are the ratios O/C and H/C, respectively. Thus, lipids are characterized by low O/C ratio and H/C ratio close to 2, carbohydrate derivatives by high O/C and H/C ratios, and lignin pyrolytic products by H/C value close to 1 and O/C ratio in the 0.2 to 0.6 range. For compounds containing nitrogen and/or sulfur atoms, N/C and S/C ratios can also be used to draw similar diagrams [30]. Regarding Kendrick’s maps, the method consists of using the mass defects in modifying the mass scale by taking, for example, CH_2_ as the new mass scale reference. Each signal associated with this new so-called Kendrick mass (KM(CH_2_)) is plotted as a function of its Kendrick mass defect relative to its nominal mass. Thus, compounds differing by one or more CH_2_ units have the same Kendrick mass defect (Figure 1C), so they all line up on a horizontal line [31].

Such graphs were helpful to follow-up oil or bio-oil upgrading treatment (elimination of oxygen, nitrogen, or sulfur by hydrotreatment or cracking) [32,33].

As reported in the different reviews [19,20,21,22,23,24,25,26], most of the HRMS analyses are performed in direct infusion (DI) mode, which may be responsible for the ion suppression phenomenon [34], especially in the electrospray ionization mode. In addition, the majority of the analyses are carried out by FT-ICR MS due to its greatest capabilities in terms of resolving power and mass accuracy measurement [18] (Table 2).

The petroleomic approach, in DI mode, shows some limitations, as only the isobaric level is well described whereas quantitative, isomeric, chemical functions, and structural information is missing. Although only a small number of research projects, mostly in the petroleum industry, have been reported, it has been demonstrated that these various characteristics are important for understanding catalytic treatments [35,36]. New analytical methodologies are needed, and some of them are already under development, to achieve a more complete molecular description of bio-oils. This review covers the recent developments to meet these objectives. Moreover, future research directions will be given, especially around the ability to obtain quantitative or at least semi-quantitative information.

Figure 2 gives a framework for this review. First, the necessity of using complementary ionization sources for the extensive description of these complex matrices will be discussed, followed by discussion of the need and the advantages to implement fractionation and separation methods prior to mass spectrometry analyses. Then will be reported the different works recently published to get a chemical and structural characterization of oil and bio-oil components.

## 2. Ion Sources Used in Petroleomics and Relation with the Compound Properties

Several ion sources are commonly used to study oil and bio-oil samples by HRMS: electrospray ionization (ESI), atmospheric pressure chemical ionization (APCI) and photoionization (APPI), and laser desorption/ionization (LDI) [37,38]. The specificities of each of these ionization sources led to the suggestion of the potential functional characteristics of the detected compounds, which can provide additional physico-chemical information.

### 2.1. Electrospray Ionization

ESI is the main ionization source in petroleomics [16,17,27] and it can be operated in positive or negative mode. It is mainly used to ionize and detect polar compounds. In positive-ion mode, basic compounds are specifically detected, especially using acids (formic acid, acetic acid) as additives [39] to promote the formation of protonated molecular ions [M+H]^+^. The negative detection mode is more adapted to the detection of acidic or neutral species by deprotonation using dopants as ammonium hydroxide to form [M-H]^−^ ions [17,34,40,41].

Other additives could also be used to promote ionization. Hertzog et al. used sodium acetate (AcONa) to promote the cationization of carbohydrate-like compounds (molecules containing a great oxygen content >5) in Miscanthus pyrolysis bio-oils by Lewis’s acid-base reaction between Na^+^ and the lone pair of the oxygen atoms [17]. The formation of such cationic adducts allows access to an extended composition description of a lignocellulosic bio-oil whereas formic acid and ammonium salts mainly promote the ionization of basic nitrogen-containing species [17]. The same methodology was used to investigate fast pyrolysis of oak bio-oil in positive ion ESI [42,43]. With AcONa, the oak sugar derivatives were sensitively evidenced by (+) ESI [43]. Mase et al. applied the same strategy to successfully investigate pinewood bio-oils [42] by (+) ESI FT-ICR MS.

Alternatively, Alsbou and Helleur studied bio-oils produced from forest residue, cellulose, and hardwood lignin (the last two were considered as reference bio-oils) by ESI-Ion-Trap MS. They used NaCl, NH_4_Cl, and formic acid for positive ion ESI and NaOH, and NH_4_Cl for negative ESI analysis [44]. According to these authors, adding NH_4_Cl can help to distinguish carbohydrate-derived products from other bio-oils components. These authors demonstrated a significant sensitivity increase by using NaOH and NaCl additives to promote the detection of deprotonated or sodium adducts, by (−) ESI and positive ion (+) ESI, respectively [44].

Other dopants were employed to analyze crude oils in greater depth, substances which could be relevant to the investigation of bio-oils. Tetramethylammonium hydroxide (TMAH) has a stronger basicity than ammonium hydroxide NH_4_OH; this is why it is preferably used as a reagent to promote deprotonation processes to ensure (−) ESI-MS measurements [45]. Indeed, TMAH ensured the detection and the assignation of more than 30,000 elemental compositions by (−) ESI FT-ICR MS of a South American crude oil (HC, NOS, NS_2_, S_2_ classes). Compared to the use of NH_4_OH, the additional detected features (close to 25,000) are related to the weak acidic compounds. Indeed, NH_4_OH ensures the efficient deprotonation of the most acidic compound classes, such as O_2_ and SO_2_ (naphthenic and sulfinic acid), whereas TMAH also allows the O_1_ (phenol), thiols or the poorly acidic N_1_, N_1_S_1_, NO, NS_2_ and SO classes of compounds to be detected by (−) ESI [45]. The main difference between NH_4_OH and TMAH reagents relates to the detection of hydrocarbon class (HC) compounds, which are not ionized by using NH_4_OH [45,46]. The fact that TMAH, unlike ammonia, does not react with carbonyls to generate imine derivates during the ESI process is another reason for its relevance in investigating bio-oils [47].

Furthermore, silver trifluoromethylsulfonate (AgOTf) could be used as a dopant in (+) ESI analysis. Lobodin et al. suggested its interest to favor the detection of aromatic and heteroaromatic hydrocarbons in four petroleum samples from different origins [48]. Thus, the aromatic hydrocarbon (HC) and the non-to-poor polar (S_1-2_ and S_1_O_1_) sulfur-containing classes were observed as cations. The use of formic acid dopant only ensured the detection of N_1_ and N_1_S_1_ classes. The cationization by Ag^+^ of aromatic HC, S_1-2_,and S_1_O_1_ species is the result of the interaction of the π electrons with the silver Lewis acid cation to form a charge transfer complex) [48]. AgOTf may present a significant interest to promote the ionization of the low polar lignin-linked bio-oil components [48,49].

### 2.2. Atmospheric Pressure Photoionization and Chemical Ionization

To characterize a wider range of compounds, more specifically those of lower polarity, APCI and APPI have been used [50,51,52,53]. Even if the involved ionization processes are quite different, these techniques ensure the efficient ionization of poorly polar aromatic (APPI and APCI) and poorly polar aliphatic (APCI) compounds, amounts of which may be significant in bio-oils, especially after catalytic upgrading treatments. APPI can ionize a large range of aromatics, usually using a dopant (toluene or acetone), to promote proton-transfer or charge exchange reactions [44]. Both protonated and radical ions are formed, which significantly increases the number of detected *m*/*z* signals [17,39,54]. As a consequence, it is necessary to achieve very high MS performances in terms of mass resolution to resolve all contributions present on an APPI or APCI mass spectrum. Besides, APPI is a less selective ionization source compared to ESI. Ware et al. used APPI for the analysis of bio-oils obtained from pine fast pyrolysis using toluene as a dopant in positive-ion mode. The results showed more aromatic and higher fractions for polar species [55]. Indeed, bio-oil components derived from lignin were selectively ionized by APPI. Other types of bio-oils were also analyzed by positive ion APPI such as softwood and esterified softwood bio-oils using 1% of formic acid. High oxygen species with low DBE highlighted the presence of alcohols and ethers. This method allowed not only the detection of compounds with N_1_ heteroatom but also the detection of compounds including boron heteroatom that was evidenced in ESI negative analysis by Javis et al. using negative ESI on bio-oils produced from the fast pyrolysis of woody biomass [39,56].

As for APCI, Stas et al. used positive and negative ion modes for the analysis of bio-oils from spruce wood, beech wood, poplar wood, and Miscanthus produced by fast pyrolysis and ablative flash pyrolysis [57]. Compounds with one to twelve oxygen atoms and high DBE values (up to 22) were detected. The negative APCI was more sensitive to lignin degradation products than positive ion APCI and showed a high abundance of compounds with two to four oxygen atoms and DBE ranging from 5 to 7. With APCI, these authors were able to detect high mass and more unsaturated species [57]. Thus, APCI provides more comprehensive information as the features detected by APCI were also observed by combining ESI and APPI results. Indeed, aromatic and aliphatic molecules were efficiently ionized by APCI [56,57]. Similar behavior was recently reported by Mase et al. in the study of a bio-oil produced by pinewood residue pellets fast pyrolysis [42]. The components of the bio-oil produced from cellulose and hemicellulose were visible under both ESI conditions utilized (with formic acid or sodium acetate dopants) while the lignin derivatives were selectively ionized by APPI. Additionally, APCI provided the ability to identify aromatic and aliphatic compounds and allowed the ionization of the molecule families that were also recognized with ESI and APPI [42].

### 2.3. Laser Desorption-Ionization LDI

LDI is a suitable source for a solid analysis. Briefly, in LDI, the key parameter is the absorption efficiency of the photon by the investigated sample. Therefore, the species which can be easily ionized have to present a significant absorption at the wavelength of the used laser, which ensures the absorption of one or more, generally two photons [56,57]. Alternatively, the ionization may occur in the gas phase by interaction (charge or proton transfer) between co-desorbed ions and neutrals. Such a process is close to what is observed in well-known MALDI experiments. Part of the lignin-derivates, the structure of which is close to those of common MALDI matrices, assist the desorption and the ionization of the bio-oil components not absorbing at the wavelength of the laser [54]. Hertzog et al. applied positive and negative ion LDI for the analysis of bio-oils [17]. In positive ion LDI, both pyrolytic lignin and carbohydrate compounds were detected. By negative LDI, a specific distribution of pyrolytic lignin compounds was evidenced [17]. The inability of non-aromatic pyrolysis products of cellulose to absorb the laser light causes LDI to suffer from several limitations (lower sensitivity toward carbohydrate-like components, fragmentation and recombination reactions due to laser-matter interaction that leads to artefact compounds) to analyze bio-oils [58]. To overcome these limitations, several authors investigated matrix-assisted LDI (MALDI) for studying bio-oils. Compared to LDI analysis, Smith et al. [59] observed a slight improvement in signal (increase of the S/N ratio) when they used colloidal graphite as a matrix. Other MALDI matrices were also investigated without significant improvement of the S/N ratio compared to LDI results [59]. Nevertheless, Qi and Volmer demonstrated the usefulness of some MALDI matrices to investigate lignin degradation products [54]. Whereas DCTB led to close results to LDI and CHCA to ion suppression phenomenon, 2,5 DHB promoted the ionization of high mass and Sox compounds. In the investigation of vanadyl porphyrins from crude oils, Giraldo-Dávila et al. [60] used cyanophenylenevinylene (CNPV-CH_3_) as a matrix. This matrix ensured the sensitive ionization of polyaromatics, porphyrin-like compounds by electron transfer (ET). The so-called electron transfer ionization MALDI (ET-MALDI) was successfully applied to the investigation, after extraction, of vanadyl porphyrins from two heavy South American crude oils [60]. The results obtained by CNPV-CH_3_ ET-MALDI were compared with those obtained by LDI and by MALDI using 3-(4-tert-butylphenyl)-2-methyl-2-propenylidene malonotrile (DCTB). CNPV-CH_3_ demonstrated an impressive selectivity to porphyrin-like compounds. ET-MALDI with CNPV-CH_3_ may be of significant interest in investigating third-generation bio fuel produced from algae to evidence chlorophyll-like compounds. Indeed, petroporphyrins are biomarkers derived from chlorophyll [60].

### 2.4. Ion Source Complementarity

Despite the specificity of each ionization source, the combined use of two or three ion sources gives complementary information regarding the complexity of bio-oils. Hertzog et al. combined the use of ESI, APPI, and LDI for the analysis of bio-oils (Figure 3). A wide range of unsaturation levels as well as high oxygen atom counts were detected in negative ESI as well as specific sugar derivatives in positive ion ESI. Furthermore, these authors found that (+) APPI could ionize less polar species (lipids and lignin derivatives) while (−) LDI was restricted to the ionization of lignin derivatives [17]. The combination of ionization sources gives information related to the effectiveness of a catalytic hydrotreatment on a bio-oil produced from lignin pyrolysis. In this context, Olcese et al. combined negative ESI and LDI FT-ICR MS analyses to evaluate the catalytic treatment of bio-oils produced by pyrolysis [61]. They identified compounds that were catalytically converted and refractory to catalytic treatment by ESI. The former was no longer observed, while ESI still detected the latter. By LDI, they identified after catalytic treatment, more conjugated and less oxygenated species corresponding to the conversion products [61].

Finally, the specificities of each ion source lead to the combination of several of them to carry out a wider non-targeted analysis of such a complex mixture [62]. Combination of ionization sources was also shown to be relevant for other organic complex matrices, such as natural organic matter, whose molecular fingerprint may be close to the bio-oil one [63,64]. It is well known that part of the NOM comes from the degradation of lignocellulosic materials.

It is a necessary step to understand the fine chemical effect of an upgrading treatment as it was highlighted for bio-oils generated from biomass pyrolysis using different catalysts [65]. In the upgrading process of a bio-oil, the need for different ionization sources is related to the important chemical modifications (cracking and deoxygenation), which modify the polarity of its components. However, the non-targeted analysis suffers from some limitations, such as ion suppression or matrix effect, which is the reason why fractionation and/or separation steps prior to such analysis is required [34].

## 3. Chromatographic Methods

As previously described, most of the studies are performed by HRMS in direct infusion mode, which is responsible for ion suppression phenomenon and ionization competition, especially in electrospray ionization [66]. This results in fewer detected compounds. A solution to overcome this issue is to implement a separative method prior to the mass spectrometry analysis.

Several studies were performed using different on-line or off-line chromatographic methods, with different detection modes such as refractive index, ultraviolet, and mass spectrometry. A recently published review pointed out such methodological approaches [32]. Among the separative methods, there are notably high-performance thin-layer chromatography [67,68,69] and gel permeation chromatography [70,71]. This latter method could be efficiently used for the fractionation of bio-oils [32,72,73] using different types of columns and could be coupled to FT-ICR MS. Tetrahydrofuran (THF) and dimethylformamide (DMF) are the typically used solvents and low temperatures are usually applied [32,71,72,73]. Several studies using GPC on thermochemically produced bio-oils such as fast pyrolysis, catalytic fast pyrolysis, and hydrothermal liquefaction oils showed that this technique could be the most practical method to use for bio-oil fractionation, based on the molecular weight of its components [57,71,74].

Liquid chromatography (LC) is also used as part of bio-oil characterization, with different columns such as C18 [72,75] and amide [76]. The one-dimensional LC analyses allowed identifying and quantifying some species but with a limited number. Therefore, to increase peak capacity and to separate bio-oil compounds over a larger polarity range, a two-dimensional LC was carried out, where reversed phase LC (RPLC) was hyphenated to a more polar column [73,77,78]. Alternatively, size exclusion chromatography can also be used as a first-dimension separation coupled with RPLC with UV and HRMS detection. This method showed almost twice as those detected in 1D RPLC. Thus, it was considered relevant for the analysis of biomass samples [72]. Another way to improve separation orthogonality is to hyphenate RPLC to supercritical fluid chromatography (SFC). SFC was already used alone with different detection modes [79,80] and it allowed observing some isomers [79]. Two-dimensional RPLC × SFC analysis was shown to reach a higher peak capacity compared to RPLC × RPLC method, with 560 and 620 peaks, respectively [80].

Gas chromatography (GC) is also frequently used to characterize the light bio-oil fraction, which represents a small part of the bio-oil components [81]. As for LC, studies have been performed by 1D (GC) and 2D (GC × GC), the latter allowing more compounds to be detected [82,83,84].

The next step concerning the characterization of bio-oil with chromatographic methods is to hyphen them with HRMS, in order to take advantage of the high resolution and mass accuracy performances. The main issue is the loss of performances of HRMS instruments at high measurement rate especially for high mass compounds. A few studies have been already performed in which high-performance mass spectrometry was hyphenated to Orbitrap-MS [68], or RPLC to FT-ICR MS [75,85]. An advantage of the separation step prior to FT-ICR MS was clearly demonstrated with 5500 formulae obtained under the hyphenated method against 2000 in DI-FT-ICR MS [75]. This was notably explained by the reduction of the ion suppression caused by the direct infusion mode.

## 4. Fractionation by Chemical Classes

### 4.1. Fractionation Methods

Among the fractionation methods, one is widely used for crude oil or bitumen, samples to separate the compounds into Saturate, Aromatic, Resin, and Asphaltene classes, which gives its name to SARA method [86]. It is based on (1) the solubility of oil components in different solvents with different polarities before (2) separation on an alumina column. This method is used to separate the component of crude oil into four different classes: saturate (cyclon), aromatic (cyclohexane/dichloromethane), resin (methanol/toluene), and asphaltene (*n*-heptane) before analysis [86,87,88,89,90]. Overall, it can be stated that SARA fractionation helps to determine the composition of crude oils by dividing them into defined hydrocarbon classes. This fractionation method could be associated with different analytical techniques such as chromatography, NMR [89], or mass spectrometry. Combining SARA with high-resolution MS measurement helps to increase both the number of the detected compounds and the knowledge of their chemical characteristics, leading to provide a better understanding of crude oil [91,92].

Cho et al. [92] analyzed the different SARA fractions of heavy crude oil by APPI FT–ICR MS to obtain a detailed characterization. The saturated fraction appeared to be composed of fewer aromatic molecules with long or multiple alkyl chains, while in the asphaltenes fraction, peri-condensed molecules were observed. The resin fraction combined nitrogen and oxygenated aromatic compounds with short carbon chains [92]. Islam et al. [91] used the SARA fractionation followed by APPI FT–ICR MS measurement to analyze weathered and artificially photo-degraded oils. These authors were able to determine the compositional charges at the molecular level by considering the heteroatomic class behavior. In the saturate fraction, the photo-degradation led to the reduction of the S_1_ class compounds with a high DBE value. At the same time, the abundance of S_1_O_1_ class compounds with high DBE values increased in the resin fraction [91].

In addition, a supplementary step to the conventional SARA fractionation can be introduced using ethyl acetate as an additional mobile phase to isolate maltenes and obtain a second resin fraction. Santos et al. [86] analyzed this fraction by FT ICR-MS using different ionization techniques. An increase in the number of detected molecular formulas was only found in this fraction two of resin regardless of the ionization technique used. A high aromaticity that was not identified throughout the crude oil analysis was also evidenced in this fraction. However, the overall DBE/C ratios for this fraction did not change significantly compared to the other fractions [86].

Furthermore, the modification of SARA fractionation can also be done by integrating separation steps to take into account the chemical properties in addition to the solubility properties [93]. This extended-SARA fractionation (E-SARA) allows key functional groups to be identified by fractionating asphaltenes depending on their adsorption at oil-water and oil-solid interfaces. These asphaltene subfractions are reported to be a real contribution to the stabilization of a water/oil emulsion [93].

These different SARA fractionation methods were widely used on petrol and could also be applied to bio-oils or formulated materials including bio-oils. Li et al. [94] applied this technique for the characterization of aged bitumen modified by bio-oils. These authors studied SARA fractionation on two types of bio-oils: pyrolysis wood PW and co-pyrolysis of wood and rubber CPWR. The latter one showed high aromatic and asphaltene fractions while the former one was dominated by the resin fraction [94]. SARA fractionation cannot be directly applied to bio-oil due to the high polarity range of its constituents. Indeed, this method must be adapted to fractionate more efficiently polar compounds with an extended scale of polar solvents. Recently, bio-oil fractionation was performed using a flash chromatography technique with a silica cartridge [95] (Figure 4). Three eluents (toluene, 90/10 toluene/methanol, 75/25 toluene/methanol, and methanol/water/formic acid) were used to obtain four distinct fractions, which were investigated by gravimetry, chromatography, and FT-ICR MS. Thus, each fraction was characterized by a specific class of compounds (fraction one, low oxygenated and unsaturated compounds; fraction two, more oxygenated unsaturated compounds; fraction three, cellulose and hemicellulose derivatives; and fraction four, the most polar compounds) allowing a better description of the molecular composition of bio-oil [95].

Furthermore, acid/base/neutral fractionation could also be used on bio-oils and was applied by Farrapeira et al. [96]. This fractionation showed the capacity to obtain four fractions of bio-oils: two acidic, one neutral, and one basic. These fractions were analyzed by 2D GC–TOF-MS. The neutral fraction was mainly composed of aromatics, ketones, ethers, and fatty esters while the basic one was formed of hydrocarbons and alkanes. Among these four fractions, the acidic one was shown to have a similar composition to bio-oil and appeared to contain high concentrations of phenol, catechol, eugenols, and furfural [96]. These results emphasized using bio-oils as a precursor in the chemical industry [97].

### 4.2. Distillation, Precipitation and Fractionation Method (DPF) Mass Spectrometry

The distillation, precipitation, and fractionation (DPF) method consists of fractionating oil into six different chemical classes before their analysis by high-resolution mass spectrometry. The six classes are volatile saturated hydrocarbons, asphaltenes, heavily saturated hydrocarbons, alkyl aromatic hydrocarbons, heteroaromatic compounds, and polar compounds [98,99,100]. The DPF procedure is described in Figure 5.

The volatile saturated hydrocarbons are obtained using vacuum distillation at room temperature. As in SARA fractionation, asphaltenes are obtained by precipitation using *n*-heptane. The *n*-heptane soluble fraction (maltenes) is divided into three compounds classes on a normal-phase silica column with respect to their elution by *n*-hexane (heavily saturated hydrocarbons and alkyl aromatic hydrocarbons), dichloromethane (heteroaromatic compounds), and finally isopropyl alcohol (polar compounds). The separation of heavily saturated hydrocarbons from alkyl aromatic hydrocarbons is conducted by solid-phase extraction [98].

In the work of Niyonsaba et al. [101], the six fractions were analyzed by APCI Orbitrap and high-resolution two-dimensional gas chromatography coupled with electron ionization high-resolution time-of-flight mass spectrometer (GC x GC TOF–MS) to achieve the characterization of each fraction at the molecular level [99,101]. APCI Orbitrap was used in direct infusion for the investigation of the non-volatile fractions in positive ion detection mode whereas volatile saturated hydrocarbons were investigated by GC x GC TOF–MS. Table 3 reports the instrumental parameters employed to analyze each of the non-volatile fractions by APCI Orbitrap [98,101].

Nitrogen is more frequently used as sheath and auxiliary gas. However, when nitrogen is used, linear saturated hydrocarbons create both [M]^+^* and [M-H]^+^ ions and undergo considerable fragmentation. To overcome this problem, some studies showed the usefulness of oxygen. When this latter is applied as sheath and auxiliary gas, only [M-H]^+^ ions are formed. No ozone or NOx formation was reported when O_2_ was used. [102,103] The only thing that should be considered when using O_2_ is the temperature, for safety reasons.

The results of the DPF MS approach were used to establish some correlation between the chemical compositions of the crude oil samples and their API gravity (American Petroleum Institute gravity) [98,99,101].

Niyonsaba et al. used DPF MS to determine the chemical compositions of five crude oil samples (two heavy from San Joaquin Valley and California, two medium from Russia and South America, and one light from Illinois) with different API gravities [101]. In addition to these methods, Ravikrian et al. used infrared spectroscopy to select tracer compounds representative of each fraction [101]. The DPF method excels in providing an accurate analysis of the crude oil (average molecular weight, heteroatom content, etc.) by including the mass balance of each fraction.

The use of this type of approach has not yet been reported in the context of the analysis of bio-oils. However, it awakens interest within the framework of the study of bio-oils improved by catalytic treatment and in that of bio-oils produced by catalytic pyrolysis. Indeed, in these two types of bio-oils, purely hydrocarbon compounds are present. Being able to distinguish heavily saturated compounds, alkylated aromatic compounds, heteroaromatic compounds, and polar compounds could allow better adaptation to the catalytic conditions in order to efficiently produce hydrocarbon compounds with higher added value.

## 5. Getting Chemical Characteristics beyond the Isobaric Level

### Ion Mobility Spectrometry IMS

Petroleomics has revealed that species present in complex mixtures such as bio-oils or crude oils present a large diversity of sulfur-, nitrogen-, and oxygen-containing chemical functions. Therefore, a raw formula, which is the only information obtained by mass spectrometry, can relate to many isomers. In this context, the association of ion mobility with mass spectrometry can be helpful to at least highlight the isomeric diversity and, in the most favorable situations, discriminate these isomers [104,105].

Ion mobility spectrometry (IMS) separates the ions through a gas-filled mobility cell under an electric field. Depending on the type of ion mobility technology, the distinction/separation of isomeric/isobaric ions according to their mass, charge, size, and shape can be achieved using different instrumental parameters. The size and shape of an ion are linked to its collision cross-section (CCS). The use of a drift tube ensures the direct correlation of the mobility of an ion with its CCS, which is an intrinsic property. To obtain the ion CCS, the other IM technologies require the use of standard compounds for calibration. The most important issue is to use a standard, in which mobility behavior is close to those of the studied compounds. Comparison of the measured CCS with the CCS calculated by molecular modeling for putative isomers, it is possible to propose detailed structural information about the analyte ion [106,107,108,109]. The IMS could be coupled to mass spectrometry IM-MS (ion mobility mass spectrometry). Therefore, before being analyzed by MS, the ions are discriminated according to their mobility in the mobility device. In addition to the *m*/*z* and the CCS descriptor, a third one may be used when tandem mass spectrometry (MS/MS) is implemented. Different ion mobility techniques that were applied to crude oils. Figure 6 gathers a selection of publications dealing with the use of ion mobility mass spectrometry in the field of petroleomics and Figure 7 highlights the advantages of the different ion mobility techniques.

Lalli et al. [105] combined IM-TOF-MS and MS/MS by collision-induced dissociation (CID) to investigate emulsion interfacial material isolated from Athabasca bitumen and heavy crude oils. These authors focused on the functional isomers from the O_3_S_1_ heteroatom class [105]. The results obtained by these authors were compared to the results obtained from the analysis of the same samples by negative ESI FT-ICR MS. It appeared that IM-TOF MS was capable of increasing the description of O_3_S_1_ heteroatom class isomers, and FT-ICR MS allowed peak assignment to be confirmed [105].

Even if it is still difficult to obtain the real CCS from the mobilogram with the main part of commercial instruments (only one is relative to a drift tube) due to the lack of a representative standard to ensure a confident calibration, the shape of the IM peak may be used to investigate the isomeric diversity. Thus, Farenc et al. [110] proposed a new descriptor to account for the isomeric content of oil-linked complex mixtures. This descriptor is based on the full width at half maximum (FWHM) of the extracted ion mobility peak [107]. These authors were able to investigate the isomeric content of N_1_ compounds with respect to the DBE and the carbon number value in vacuum gasoline oil (VGO).

Both previous works were performed with a high-resolution TOF mass spectrometer. This instrument suffers from a lack of resolving power, which does not always ensure resolution of isobaric species when complex mixtures are investigated. To overcome these limitations, Rüger et al. [117] and Cho et al. [116] explored recently the capabilities of cyclic ion mobility high-resolution mass spectrometry (cIMS) in the field of petroleomics. Quadrupole-selected cyclic ion mobility mass spectrometry (QcIMS) ensures transfer of ions of a given *m*/*z* range into the cIMS device. The number of cycles (passes) in the cyclic IMS may be increased to ensure efficient separation of isomeric ions. Rüger et al. [117] succeeded to investigate the PAH and PASH composition of VGO by the combination of IMS and MS and overcoming the sulfur split (SH4 and C3 isobaric interference) [117]. Moreover, the use of MS/MS allowed structural groups of molecular core isomers (PAH and PASH with different alkylation degrees) to be distinguished. A similar approach was used by Cho et al. to investigate crude oils [116]. 

More recently, Maillard et al. [119] used for the first time the combination of trapped ion mobility (TIMS) and FT–ICR MS to obtain information on the structural features and isomeric diversity of vanadium petroporphyrins in heavy petroleum fraction (Figure 7) [119]. The determination of the CCS and its correlation with theoretical calculation allowed them to propose some putative structures and the use of the FWHM descriptor ensured the accounting for isomeric diversity.

Regarding IM-MS analysis on bio-oil, a few studies were reported in Figure 6. Among these studies, Dhungana et al. applied IM-MS on bio-oils from biomass pyrolysis. As a result of the discovery of oxygen-rich species (one and nine oxygen atoms) with DBEs ranging from 1 to 15, it is anticipated that catalytic upgrading will be necessary if slow-pyrolysis bio-oils are to be used as fuel. IM-MS/MS gives complementary information related especially to isomer separation in addition to the traditionally used HRMS in the analysis of bio-oils [115].

However, ion mobility spectrometry still suffers from a lack of mobility resolving power. Figure 8 highlight the different advantages of IMS technologies. Dodds et al. investigated the correlation between mobility resolving power (mRP), and CCS. These authors demonstrated that the most advanced ion mobility (IM) instruments reach now mRP more than 300 and are therefore capable of separating compounds with CCS differences as low as 0.5%. Current IM instruments operate across a broad range of separation efficiencies between 50 and 300 mRP (CCS/ΔCCS). It would take resolving powers on the order of several thousand to resolve the majority of the components in a complex mixture using ion mobility, which is far beyond the capabilities of current instrumentation due to peak broadening limits imposed by ion diffusion [120].

## 6. Tandem Mass Spectrometry

Tandem mass spectrometry is a well-known technique to obtain structural information. Typically, the parent (also named precursor ion) is isolated prior to increasing its internal energy. This leads to the dissociation of part of the chemical bonds to produce fragments also named daughter ions. The activation of the parent ion may be performed by vibrational excitation though absorption of infrared photons (the so-called infrared multiphoton dissociation (IRMPD)) or by collision with a gas (collision induced dissociation CID) [124,125,126,127,128,129,130,131]. Alternatively, electronic activation may be performed by the absorption of a UV or VUV photon (UV photodissociation UVPD). Another way to induce the fragmentation of parent ion is to generate a radical by interaction with an electron observed in electron capture dissociation (ECD), electron transfer dissociation (ETD) and electron induced dissociation (EID) experiments [132,133,134]. Although this latter activation method can be applied to singly charged ions, it is mainly used to study multi-charged ions. In petroleomics, the most frequently used activation techniques are CID and IRMPD.

Among the different applications of tandem mass spectrometry in petroleomics, this section will be more specifically dedicated to bio-oils, especially the one produced from lignocellulosic biomass. It is widely known that this raw material is made of biopolymers, namely cellulose, hemicellulose, and lignin, making it a promising source. In order to determine their structure, tandem mass spectrometry has been used, and many investigations are reported.

### 6.1. Carbohydrates

The most prevalent kind of carbohydrates in lignocellulosic biomass is cellulose, a glucose polymer and hemicellulose [135]. Hemicellulose is made of C6- (glucose, mannose, galactose) and C5- (xylose and arabinose) carbohydrate units [136]. The structural elucidation of isomeric sugars by MS/MS is very challenging. The main part of the investigation by MS/MS conducted on bio-oil carbohydrate component did not lead to a complete structural elucidation but rather to the confirmation of the presence of such compounds in bio-oils. Vinueza et al. demonstrated the use of chloride anion attachment in APCI and ESI tandem mass spectrometry for molecular weight MW measurement and for a partial structural elucidation of various mono-, di-, and oligo-saccharides without sample preparation or derivatization [137]. Significant HCl loss (chlorine-containing fragment anions) was seen in the collision-induced dissociation CID of the carbohydrate chloride adducts anion. This particular loss was utilized not only to prove that the fragmenting ion was a chloride anion adduct but also it allowed the distinction between carbohydrates containing no free anomeric hydroxyl groups and those that contain them [137]. In addition, Yu et al. used a similar approach to determine the first reactions and byproducts of fast pyrolysis of xylobiose and xylotriose, model xylan compounds [138]. Quantum chemical computations were used by these authors to study the processes of the ring opening and subsequent fragmentations of the reducing end. In particular, it was shown that ring-opening and coordinated removal of ethenediol from the reducing end of xylobiose were necessary for the synthesis of β-D-xylopyranosylglyceraldehyde (XGRA), the most prevalent fast pyrolysis product of xylobiose and xylotriose. It is claimed that coordinated Maccoll elimination and/or Pinacol ring contraction led to the cleavage of the glycosidic bond and 1,2-dehydration [138].

### 6.2. Lignin

Among the biopolymers mentioned previously, lignin is the second largest component of lignocellulosic biomass, accounting for 15% to 30% of the mass, and it is generally considered to be composed of three phenylpropane units [139,140,141,142]. However, tandem mass spectrometry is a promising tool for lignin structural analysis, as fragmentation patterns of model compounds can be extrapolated to identify characteristic moieties in complex samples [143]. Zhang et al. applied tandem mass spectrometry on alkali lignin pyrolysis oil after the sequential extraction method. For the raw pyrolysis oil and the subfractions, a peak at *m*/*z* 360 with a high relative abundance was observed [144]. The precursor of this ion was a dimer connected by a ferulate bond and coniferyl alcohol. Due to the absence of weak binding groups, no further fragmentations were seen in the two other investigated precursor ions [144]. In another work, Abdelaziz et al. investigated the first continuous-flow reactor setup for base-catalyzed depolymerization of lignin using sodium hydroxide to create low-molecular-mass aromatics [145]. Size exclusion chromatography (SEC), nuclear magnetic resonance spectroscopy (NMR), and supercritical fluid chromatography-diode array detector-tandem mass spectrometry (SFC-MS) were used by these authors to characterize the obtained products (phenolic bio-oil consisting of monomeric/oligomeric aromatic compounds) [145]. As a result, the aliphatic C-O bonds in the lignin inter-unit structures (β-O-4, β-β, β-1, β-5), were broken under the investigated reaction conditions, providing evidence of efficient lignin depolymerization [145]. Moreover, the current lignin depolymerization technique also made it possible to produce partially deoxygenated dimeric and oligomeric fractions for use in liquid fuel applications and other renewable energy sources [145]. Recently, Dong et al. used beam-type CID with energy-resolved mass spectrometry (ERMS) to apply previous resonance excitation type CID techniques that identified lignin oligomers containing β-O-4, β-5, and β-β bonds to additionally identify features of 5-5, β-1, and 4-O-5 dimers [143]. Overall, the beam-type ERMS provides comprehensive structural data and may eventually help develop instruments for high-throughput lignin dimer detection [143].

However, with real bio-oil samples, many mass peaks at each nominal mass render this approach inefficient, mainly due to the low resolution of quadrupole ion precursor isolation for CID. The resolution necessary for the isolation of individual ions can be achieved in the ICR cell by FT-ICR MS, but these measurements are very slow and tricky and involve a decrease in the performance of the instrument due to the gas introduced into the vacuum part for the CID. To overcome this problem, two-dimensional tandem mass spectrometry using FT-ICR MS will be helpful, especially for bio-oils. It enables the creation of two-dimensional maps of parent ions as well as fragment ions that were activated by infrared or UV photon, or by interaction with an electron (ECD, ETD or EID). The advantage of this modern approach, which is presently employed to study ionized peptides by DIA-ESI, is that it does not require the conventional parent ion selection step. In order to investigate a complex mixture, it is therefore possible to obtain all the fragmentation spectra of the ions present on a mass spectrum in a reasonable length of time, ranging from a few tens of minutes to a few hours. Several studies were reported on using IRMPD 2DMS for the analysis of polymers [129,130], peptides [131,146] and, agrochemicals [127,128].

However, to our knowledge, the bio-oil analysis by MS/MS utilizing HRMS is limited at this time, but it may be able to offer structural data on the bio-oil constituents for deeper characterization.

## 7. Specific Reactants to Chemical Functions

To go deeper into the description of structure of oil constituents at the molecular level, some approach has been proposed to evidence by derivatization specific chemical functional groups, as showed in Table 4. Highlighting these functional chemical characteristics is of great importance to understand the action of treatment of a bio-oil but also to guide their action. Most of the derivatization processes can be considered with bio-oil due to its wider molecular complexity in terms of chemical functions.

### 7.1. Carbonyls

The bio-oils and the crude oils contain wide amounts of oxygenated compounds that are related to different functional groups [57,147]. Among these oxygenated functional groups, the carbonyls (aldehydes and ketones) are of major concern because of their significant reactivity and abundance, with up to 20 wt.%, in bio-oils [147]. There are also well-known carbonyl-specific reactions. The most important ones are the formation of imines by reaction with amines or hydrazone by reaction with the 2,4 dinitrophenylhydrazine (2,4 DNPH) or the Girard T reagent [38,148,149,150,151].

Hertzog et al. [47] evidenced some carbonyl compounds by ESI positive ion FT–ICR MS in bio-oils, after derivatization with aniline and 3-chloroaniline evidence [47]. The detection of chlorine-containing species, such as C_x_H_y_O_z_NCl and C_x_H_y_O_z_N_2_Cl_2_, allowed these authors to unambiguously define both mono and di-carbonylated C_x_H_y_O_z_ compounds. The elemental formula of the original compound corresponding to these carbonyls is obtained by the subtraction of one (or two) C_6_H_4_ClNH_2_ and the addition of one (or two) H_2_O. This reaction took place in the ESI source itself [47].

In the field of corrosion tests of naphthenic acids, Krajewski et al. [152] used an alternative approach. After extraction of the ketones on a strong anion exchange (SAX) cartridge from a corrosion test naphthenic acid commercial sample, they were derivatized by the Amplifex keto reagent. The resulting compounds were analyzed by (−) ESI FT–ICR MS. The N_2_O_1_ features relative to derivatized O_1_ ketone ensured to determine the oxidation products of the naphthenic acids [152].

### 7.2. Phenols and Alcohols

Alcohols, and, more importantly, phenols, are other classes of compounds observed in bio-oils with a great amount (up to 40 wt.%) and to a lesser extent in petroleum. Their abundance is depending on the origin of the oil [153]. They can be quantified by gas chromatography mass spectrometry in complex petrochemical samples after derivatization to esters of ferrocene carboxylic acid [154]. Indeed, Wasinski and Andersson [154] used the esterification reaction between ferrocene carboxylic acid chloride and phenol and alcohol [154,155]. The introduction of an iron atom by the derivatization reagent may be a very powerful tool for investigating hydroxylated compounds in complex mixtures.

Another interesting derivatization method of phenol was proposed by Zhu et al. [156] who investigated such compounds in jet fuel [156]. These authors used an imidazolium-based charged tag (3-(4-(bromomethyl) benzyl)-1-methylimidazolium) that in situ reacted with phenol to produce an ether. This approach significantly increased the sensitivity due to the charged-tagged derivatization. Moreover, it was also demonstrated that thiols may also be derivatized by the same methodology, which ensured the distinction of thiols and thioethers. Nevertheless, classical derivatization techniques could also be applied to derivatize and quantify phenols in complex mixtures such as bio-oils. Among these classical derivatizations, acetylation with acetic anhydride in the presence of pyridine was used and applied by Murwanashyaka et al. [157]. GC-MS analysis following this pretreatment (fractionation and acetylation) was applied to monitor the yields of phenols in the obtained samples [157].

Our group is currently working on the derivatization of -OH functions of phenols, aliphatic alcohols, and carboxylic acids in bio-oil, by pentafluoropyridine and deuterated iodomethane. Indeed, iodomethane, also named methyl iodide, was already used as methylation agent for -OH and -COOH groups in natural organic matter [158,159,160]. This study will be the subject of an upcoming article.

### 7.3. Carboxylic Acids

Among the different oxygenated substances that can be present in bio-oils, carboxylic acids are one the most reactive and concerning compounds. Acidic compounds are responsible for corrosiveness issues and, therefore, storage problems of bio-oils. Moreover, they can also be involved in condensation reactions with other bio-oil compounds, which yield water and decrease the high heating value HHV value of this material and can also lead to the formation of two phases in the sample. For these reasons, carboxylic acid characterization in bio-oil is of major interest, and derivatization is a way to analyze them.

Esterification as benzyl esters is an example of these reagents and was applied by Staš et al. to characterize pyrolysis bio-oil (oak-maple, pine, and straw oils) using GC-MS. These authors showed that the amount of the derivatized acids can be influenced by the type of biomass [57,161]. However, formic and acetic acid was found to be the majority regardless of the type of biomass. Also, Staš et al. applied another modification by using a phosphitylating agent to characterize and quantify acids of the bio-oils. The phosphitylating agents that could be used: the 2-chloro-4,4,5,5-tetramethyl-1,3,2-dioxaphospholane (TMDP) and 2-chloro-1,3,2-dioxaphospholane [57,161,162]. The former was shown to provide a better signal than the latter. By employing ^31^P NMR, these P-agents could quantify the acids in bio-oil [163,164]. Nevertheless, other phosphorylation reagents exist and were applied by Balakshin et al. [163] for the analysis of carboxyl groups: 1,2,3-dioxaphosphalane and 2-chloro-4,4,5,5-tetramethyl-1,3,2-dioxaphospholane. As evidenced by the authors, the second result showed a good correlation with those from ^13^C NMR after quantification with ^31^P NMR [162,163,164].

Another derivatization reagent exists to modify carboxylic acids such as methylation by esterification reaction using diazomethane or BF_3_/methanol and trimethylsilyltion using the *N*,*O*-bis(trimethylsilyl) trifluoroacetamide (BSTFA) [164].These reactions usually happen by replacing the active hydrogen atom of the carboxylic acids with a methyl or trimethylsilyl group forming carboxylic acids esters and trimethylsilyl ethers, respectively.

**Table 4 molecules-27-08889-t004:** Examples of derivatization reagent used to modify oxygenated functional groups in bio-oils.

Functional Group	Derivatization Reagent	Sample	References
Carbonyls	2,4 Dinitophenylhydrazine (2,4 DNPH)	Petroleum resin and Asphaltene	[47,148,149]
	Girard T	Crude oils	[38,151,165]
	Quaternary aminoxy (QAO)		[151]
	Aniline	Bio-oils	[47,166]
	3-Chloroanaline	Bio-oils	
	Amplifex keto	Naphthenic acids	[152]
Phenols and alcohol	Ferrocene carboxylic acid chloride	Pyrolysis oil and petrochemical sample	
	(3-(4-(Bromomethyl) benzyl)-1-methylimidazolium)	Jet fuel	[73,154,155]
	Acetic anhydride	One-step and stepwise laboratory batch vacuum pyrolysis of a mixture of birch bark and birch sapwood	[156]
	Butyl chloroformate	Fuels and engine oils	[157]
	Pentafluoropyridine	Bio-oils	[167]
Carboxylic acid	Alkylation (benzyl ester)	oak-maple, pine and straw oils	[57,161,163,164]
	Tetramethyl-p-phenylenediamine (TMPD)	Pyrolysis bio-oil	
	2-chloro-1,3,2-dioxaphospholane		
	1,2,3-Dioxaphosphalane		
	2-chloro-4,4,5,tetramethyl-1,3,2-dioxaphospholane		
	Diazomethane	Bio-Oils Derived from Lignocellulosic Biomass	[147,168]]
	BF_3_/methanol		
	BSTFA		
	Tetramethylammonium acetate	Humins	[169]

## 8. Isotopic Labelling

Besides the previously described methodologies to obtain structural information, another one has emerged recently in the field of petroleomics. It is the hydrogen/deuterium H/D and the ^16^O/^18^O exchanges (Figure 9). It is a well-established methodology to investigate a wide range of organic and biological compounds by mass spectrometry. More specifically, H/D exchange (HDX) is used in proteomics to examine the conformation of proteins or in tandem mass spectrometry to confirm and understand the fragmentation pathway. Combined with high-resolution mass spectrometry, the HDX approach ensures the obtaining of chemical and structural information about an individual component of a complex mixture. Isotope labeling was already performed for the analysis of natural complex mixtures [170,171]. The capabilities of H/D exchange in mass spectrometry have been recently reviewed by Kostyukevich et al. [172] Typically, the H/D experiment is performed in solution, and exchangeable hydrogen atoms are only the labile ones. Recently, Nikolaev group developed a new methodology, which ensures the online H/D exchange reaction in the ionization source under atmospheric pressure. In that case, the labile hydrogen atoms are exchanged and part of the less or non-labile ones [173]. This approach has been used in various applications and more specifically in the petroleomic field to investigate oils and bio-oils, whatever the atmospheric ionization source (ESI, APCI, and APPI). After the dissolution of a vacuum residue in a mixture of toluene-d8 and methanol-d4 and the addition of D_2_O and DCOOD to increase the efficiency of HDX. Zhang et al. investigated the compositional variation of the intermediate amine compounds during the heavy petroleum hydrotreatment process for N_1_ compounds [174]. Depending on the number of exchanged labile hydrogen atoms, they were able to evidence pyridines (N_1_D_1_), cyclic (N_1_D_2_), and primary (N_1_D_3_) amines by FT-ICR and Orbitrap MS. The observation of N_1_D_2_ (not detected in the feedstock) in the treated heavy petroleum nicely demonstrated the conversion of pyridines into cyclic amines. The application of this methodology to the effluent of four-reactor pilot hydrotreatment plants allowed the authors to compare the efficiency of the HDN. The use of a similar approach by Cho and al. ensured the investigation of the *N*-containing class compound in crude oils by APPI HDX mass spectrometry. These authors confirmed that the prominent part of N_1_ crude oil components was pyridines [175]. Moreover, these authors applied this methodology to the resin fractions obtained by the SARA fractionation of two oils. More interestingly, this methodology was employed to study the compositional changes at the molecular level of weathered oils in the context of the Hebei Spirit oil spill [176]. The comparison of the resin fraction of the crude oil and the associated weathered oils by APPI HDX mass spectrometry led the authors to understand that secondary and tertiary N_1_ amine-containing compounds were preferentially degraded during the early stage of the natural weathering. In the alternative methodology developed by the Nikolaiev group, in-ESI source isotope exchange was conducted. The atmosphere was saturated by D_2_O vapors in the region between the ESI tip and the inlet of the desolvating capillary by placing the D_2_O drop on a copper plate close to the ESI needle. In these conditions, both labile hydrogen and non-labile hydrogen atoms were exchanged due to gas-phase reactions. The proof of concept of this approach and the mechanism/kinetic description had been extensively described in a recent paper [173]. Acter et al. [177] used a similar methodology to investigate nitrogen-, oxygen- and sulfur-containing compounds in the polar fraction of petroleum samples by in-APPI HDX mass spectrometry. In this work, the D_2_O was continuously delivered on a metallic plate in the APPI source. For part of the S_1_ compounds, these experiments evidenced the presence of one exchangeable hydrogen atom attached to the sulfur atom (thiol), whereas two H/D exchanges were observed with S_2_O_1_. Thus, comparable results were observed using HDX protocols in source APPI HDX mass spectrometry. To go deeper into the structural description of the crude oil components, Kostyukevich used “advanced” isotope exchange in near-critical water using both D_2_O and H_2_^18^O [178]. H/D or ^16^O/^18^O may be performed in near-critical water conditions by sealing crude oil with added D_2_O or H_2_^18^O in an autoclave at a temperature greater than 573 K and a pressure greater than 30 MPa (the critical temperature and critical pressure of water are 647 K and 22 MPa, respectively). H/D exchange was observed in these experimental conditions for labile hydrogen atoms from functional groups such as –OH, –COOH, –NH, and –SH and, for non-labile hydrogen atoms such as aromatic hydrogens or alpha hydrogens. ^16^O/^18^O exchange occurred for the oxygen of the carbonyl group. Up to three ^16^O/^18^O exchanges have been observed in the investigated crude oil. The authors determined that ^16^O/^18^O exchanges occurred for 276 of the 1500 detected features observed by negative ESI mass spectrometry. The combination of these isotope exchanges with NMR measurements led the authors to propose some putative structures [179]. More recently, the same authors applied the same approach and used H/D or ^16^O/^18^O isotopic exchange to investigate bio-oils produced by HTL of different raw materials including biomass and food waste [179]. By combining isotope exchange, which gained insights about the presence of functional groups in an individual molecule, tandem mass spectrometry, and specific software for the data treatment, they proposed a putative structure for close to 350 individual features.

## 9. Towards a Quantitative Approach to Petroleomic Data

Since ionization efficiency is highly component-dependent, notably for ESI, quantification by untargeted analysis is critical. However, as demonstrated by Son et al. [180]. APPI FT-ICR MS measurements of four crude oil samples over 3 months showed very good repeatability, which was the first step for consistent semi-quantitative data. As described in previous sections, ionization competition increases with the complexity of the sample. Some solutions have been proposed in the literature to move towards quantitative data or at least semi-quantitative data. The first consists of reducing sample complexity by using a fractionation or separation step prior to mass spectrometry analysis. For example, Santos et al. used the SARA methodology to obtain light compounds, asphaltene, saturate, aromatic, and two resin fractions before positive ESI or APPI analysis [86]. Rodgers et al. used a similar approach by considering the modified aminopropyl silica (MAPS) fractionation of bitumen [181]. Lacroix-Andrivet et al. used another methodology by combining high-performance thin-layer chromatography (HPTLC) and laser desorption-ionization (LDI) FT-ICR MS to analyze bitumen [182]. The HPTLC of bitumen on cellulose with a heptane/ethanol solvent mixture ensured the separation of maltenes and asphaltenes. The LDI provided the direct analysis of both fractions. The chemical properties of the compounds in the obtained fractions were generally thought to be close enough to reduce the ionization competition phenomenon.

The second approach combined high-resolution mass spectrometry and quantitative comprehensive GCxGC analysis. Guillemant et al. investigated the N_1_ compounds of different gas oils by GCxGC nitrogen chemiluminescent detection (NCD) and positive or negative ESI FT-ICR MS [183]. The studied gas oils underwent a Flash HPLC pre-fractionation before the analysis to remove the hydrocarbon matrix. The NCD ensured the absolute quantitation of nitrogen compounds. Indeed, the measured signal was directly proportional to the concentration of the nitrogen amounts. The comprehensive GCxGC-NCD led to the efficient separation of basic and neutral N_1_ compounds. Moreover, it was possible to distinguish on the 2D chromatogram, for each N_1_ class, the compounds with respect to their DBE value. Consequently, the absolute concentration of indoles, carbazoles, tetrahydroquinolines, anilines, pyridines, quinolines, and acridine was obtained. The obtained concentration by GCxGC-NCD did not linearly correlate with the integration of the signal observed by ESI in positive (tetrahydroquinolines, anilines, pyridines DBE 4-5-6, quinolines 7-8-9, and acridine 10-11-12) and negative (indoles DBE 6-7-8, and carbazoles DBE 9-10-11) detection modes. Thus, the ionization competition phenomenon was evidenced. Consequently, the concentration of a given compound and its abundance on the mass spectrum were not simply correlated. The use of multiple linear regression (MLR) ensured the establishment of such correlation. Interestingly this MLR procedure also ensures the evaluation of the ionization competition phenomenon. Alternatively, Reymond et al. proposed an off-line comprehensive three-dimensional method to quantify polycyclic aromatic hydrocarbons (PAH) in vacuum gas oils [184]. After centrifugal partition chromatography (CPC) fractionation, the resulting fractions were analyzed by supercritical fluid chromatography hyphenated to APPI FT-ICR MS. The CPC separation, according to alkylation degree, limited the ion suppression phenomenon. In the used experimental conditions, the MS area of PAH and heavy PAH correlated with their concentration in the investigated gas oils. This approach was validated by spiking the analyzed samples with pyrene-d10. The same authors used this methodology to monitor the evolution of PAH concentration during the hydrotreatment of different vacuum gas oil feedstocks [185].

## 10. Conclusions

According to the great interest in generating valuable products from biomass conversion, petroleomic-based methodologies play a key role in deciphering the complexity of these products and guiding the chemical conversion. From the high-resolution mass spectrometry data, a thorough description at the molecular level of the oligomeric composition of such products is highlighted. However, it is clear that there are compound knowledge gaps that need to be filled to achieve the goals of an accurate control of chemical conversion. It is especially crucial to get access to the chemical properties of the numerous features highlighted by HRMS to understand their reactivity. As it suggested in this review, some promising routes to bring some supplemental data on those complex mixtures can be drawn, mainly based on what is developed in the field of petroleum. Recent developments now attempt to reduce the experimental limitations such as selective ionization by including fractionation and separation methodologies before the mass analysis. Additionally, the description of the diversity of compounds giving rise to a MS signal in petroleomics can be explored in different ways. Ion mobility mass spectrometry is promising regarding the isomeric level but, to this day, only a few studies have been reported on the bio-oils, showing the great research opportunities. Several new approaches integrate tags of chemical characteristics by classes or more precisely by the functional groups involving derivatization and labelling steps. This aspect is also poorly studied in the field of bio-oil characterization and deserves to be integrated into analytical workflows.

## Figures and Tables

**Figure 1 molecules-27-08889-f001:**
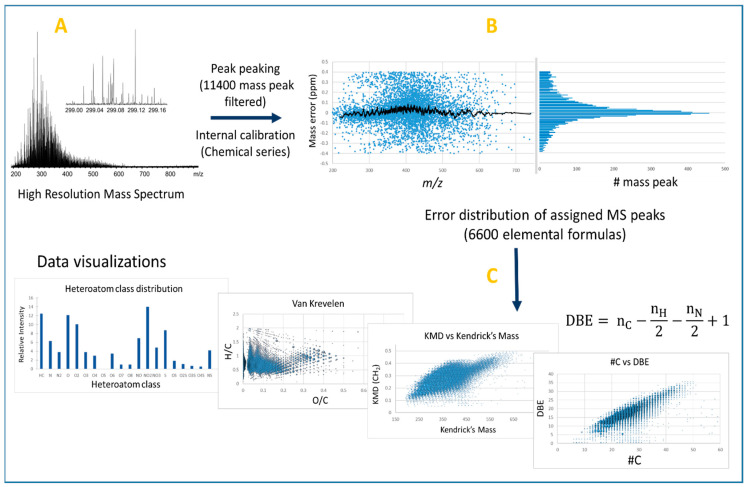
Typical petroleomic workflow: (**A**) acquisition of mass spectrum in high resolution mode (here LDI of coal-type sample); (**B**) distribution of mass measurement errors of assigned peaks after internal calibration of HR mass spectrum—the histogram is used to verify the normality of the error distribution; and (**C**) description of sample composition by various visualization tools.

**Figure 2 molecules-27-08889-f002:**
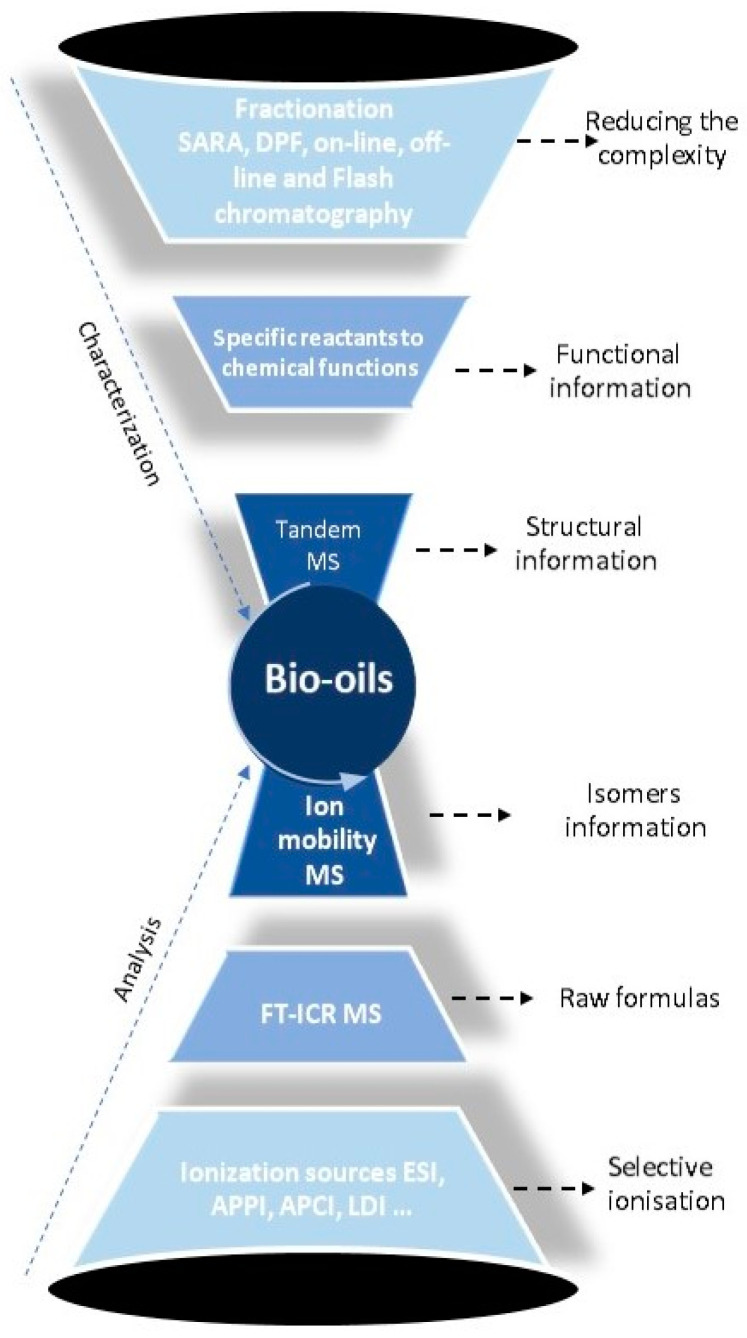
New methodologies in petroleomics that contribute to the chemical characterization of bio-oil at the molecular level.

**Figure 3 molecules-27-08889-f003:**
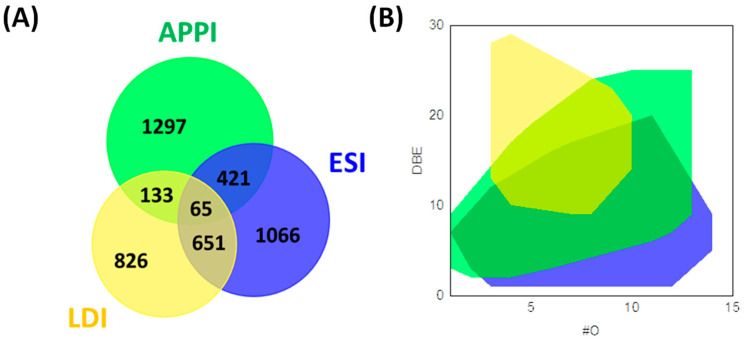
Venn diagram (**A**) and DBE vs. #O (**B**) of the CHO formulae obtained from ESI, APPI, and LDI FT-ICR MS analyses. The diagram was reprinted with permission from Ref. [17]. **2017**, *Elsevier*.

**Figure 4 molecules-27-08889-f004:**
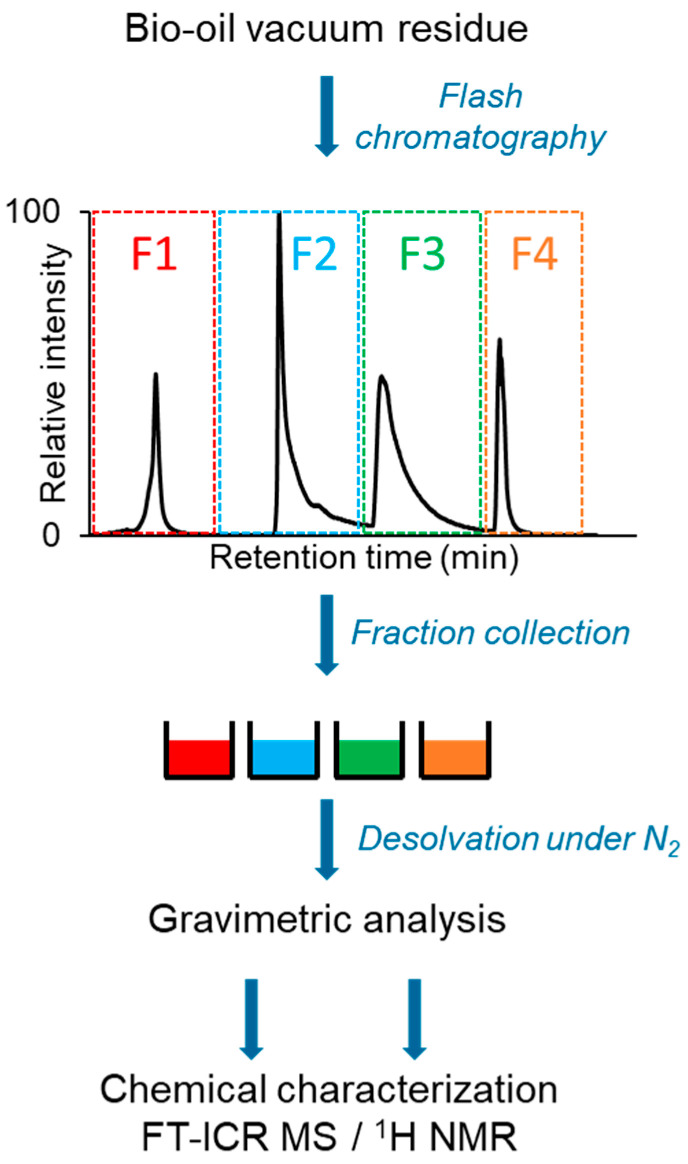
Scheme of the experimental workflow from sample fractionation to gravimetric analysis and molecular characterization. The diagram was reprinted with permission from Ref. [95]. **2022**, *Elsevier*.

**Figure 5 molecules-27-08889-f005:**
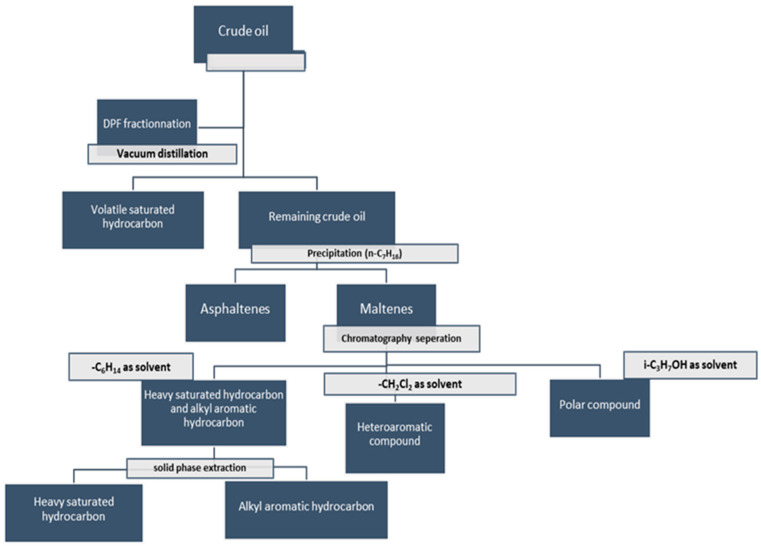
DPF fractionation steps [98].

**Figure 6 molecules-27-08889-f006:**
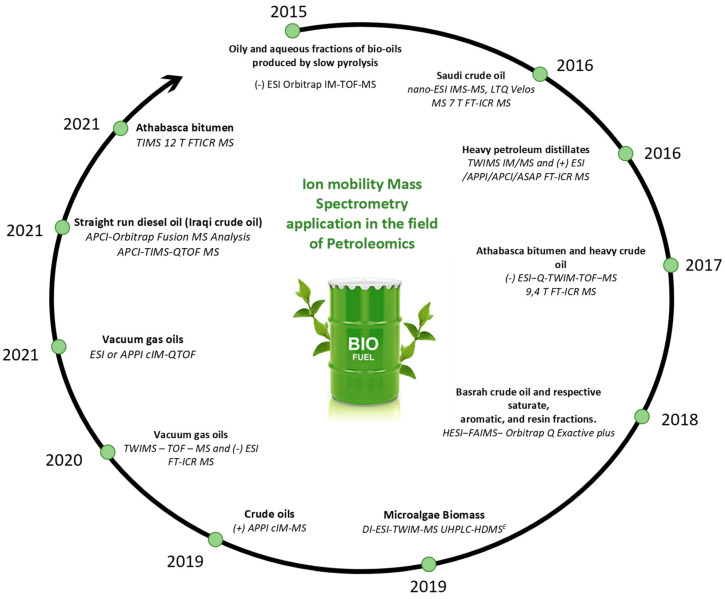
Main publications of IM-MS petroleomics [105,110,111,112,113,114,115,116,117,118,119].

**Figure 7 molecules-27-08889-f007:**
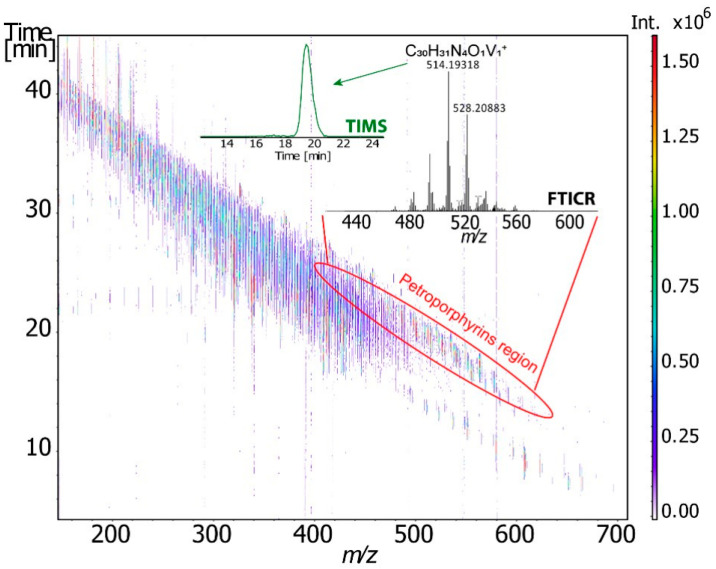
2D TIMS-FT-ICR-MS survey spectra of the Athabasca asphalten. Reprinted with permission from Ref. [119]. **2021**, *Carlos Afonso*.

**Figure 8 molecules-27-08889-f008:**
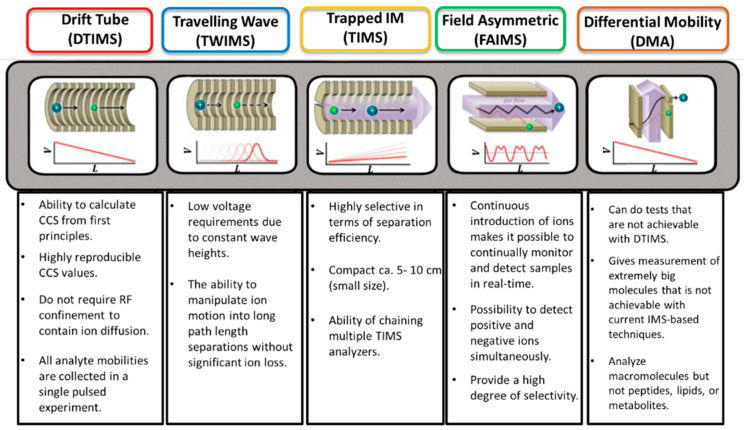
Advantages of the different of IMS technology [108,121,122,123]. Instrument diagram reprinted with permission from Ref. [123]. **2011**, *Elsevier*.

**Figure 9 molecules-27-08889-f009:**
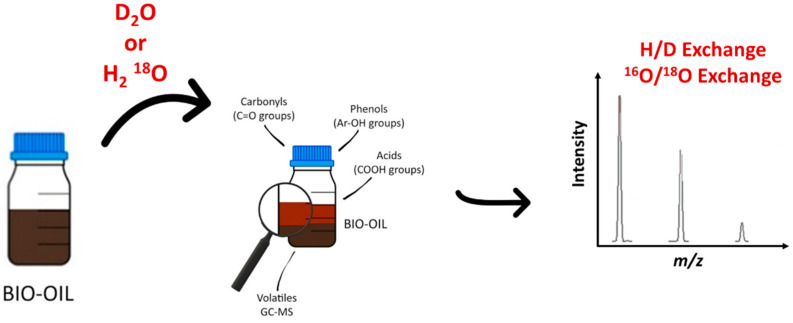
The outline for the isotope labelling.

**Table 1 molecules-27-08889-t001:** Typical physicochemical properties of heavy, 2G pyrolysis, 2G liquefaction, and 3G oil [9,10,11].

Properties		Heavy Oil	Pyrolysis Bio-Oil		Liquefaction Oil (Corn Straw, 2G)	Pyrolysis of Aegle Marmelos (3G)
			Wood (2G)	Straw (2G)		
*Elemental composition*, wt. %	*C*	85	54–58	48.93	44.57	40.0497
	*H*	11	5.5–7.0	6.52	5.53	6.1474
	*O*	1	35–40	42.56	33.70	52.2853
	*N*	0.3	0–0.02	1.11	0.93	1.4732
	*S*	1.0–1.8	0.0–0.02	0.0–0.06	0.10	0.0005
	*Ash*	0.1	0–0.2	5.45	7	2.19
*Water content*, wt. %		0.1	15–30	6.5	7–15	3.01
*Higher heating value*, MJ·kg^−1^		40	16–19	18.01–23.3	16.96	20.17

**Table 2 molecules-27-08889-t002:** Performances of several HRMS instruments (manufacturer’s specification sheet). An asterisk (*) indicates maximum resolving power (the resolution decreases as the acquisition measurement rate increases). (ND: No Data.)

Instrument	Resolution	Error (ppm, Internal Calibration)	Sensitivity (ESI)	Typical Measurement Rate	Mass Range (*m*/*z*)	Instrument Model
Orbitrap	1 M * at *m*/*z* 200	<1 ppm	100 fg reserpine (S/NL 100:1)	1 Hz	50–8000	Eclipse tribrid ThermoFisher Scientific
FTICR	>20 M * at *m*/*z* 400	<600 ppb	100 amol Ubiquitin (S/N > 20:1)	0.5 Hz	100–10,000	7 Tesla 2XR Bruker
Q-TOF	60,000 at *m*/*z* 1222	<800 ppb	100 fg reserpine (S/N: 100:1)	50 Hz	20–40,000	Impact II Bruker
timsTOF	60,000 at *m*/*z* 1222	<800 ppb	100 fg reserpine (S/N: 100:1)	50 Hz	20–>20,000	timsTOF Pro 2 Bruker
Q-Tof MRT	>200,000	<500 ppb	ND	10 Hz	50–>3000	Select Series MRT Waters

**Table 3 molecules-27-08889-t003:** Sheath and auxiliary gas and solvents used for the analysis of DPF fractions by positive ion APCI [101].

DPF Fraction	Sheath And Auxiliary Gas	Solvent
Asphaltenes	N_2_	CS_2_
Heavy saturated hydrocarbon	O_2_	*n*-Hexane
Alkyl aromatic hydrocarbon	N_2_	CS_2_
Heteroaromatic		
Polar compound	N_2_	*n*-Hexane and methanol

## Data Availability

Not available.

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
