# Peer review of "Next Challenges for the Comprehensive Molecular Characterization of Complex Organic Mixtures in the Field of Sustainable Energy"

_molecules, 2022, doi:10.3390/molecules27248889_

Round 1

Reviewer 1 Report

This work reviews the latest advances in bio-oils characterization by resorting to the non-targeted petroleomic approach. The use of high-resolution mass spectrometry techniques is described, namely FT-ICR-MS. Special emphasis is given to the ion sources used in petroleomics, such as ESI, APCI, APPI, and LDI. In section 3 the authors review chromatographic methods that can be hyphenated with HRMS, in an offline or online mode. Section 4 discusses the fractionation techniques used in the petroleomics approach, namely SARA, and their application to the characterization of bio-oils. The advantages that can be obtained using ion mobility spectroscopy and its hyphenation with HRMS techniques are discussed in section 5. The use of tandem mass spectrometry techniques is discussed in section 6, while chemical derivatization, isotopic labeling, and the approach to quantitative techniques, are discussed in sections 7, 8, and 9, respectively.

This is a very good and extensive review work, with 177 bibliographical references, which deserves to be published.

Some minor remarks:

1.      The authors should review the text carefully to remove unnecessary hyphens and to correct mistyping.

2.      The list of abbreviations should be extended to include acronyms such as CPWR, ERMS, HHV, SFC, TIMS, VGO, or VUV. In Abbreviations should be CCS instead of CSS.

Author Response

Answer and corrections - Review report

This work reviews the latest advances in bio-oils characterization by resorting to the non-targeted petroleomic approach. The use of high-resolution mass spectrometry techniques is described, namely FT-ICR-MS. Special emphasis is given to the ion sources used in petroleomics, such as ESI, APCI, APPI, and LDI. In section 3 the authors review chromatographic methods that can be hyphenated with HRMS, in an offline or online mode. Section 4 discusses the fractionation techniques used in the petroleomics approach, namely SARA, and their application to the characterization of bio-oils. The advantages that can be obtained using ion mobility spectroscopy and its hyphenation with HRMS techniques are discussed in section 5. The use of tandem mass spectrometry techniques is discussed in section 6, while chemical derivatization, isotopic labeling, and the approach to quantitative techniques, are discussed in sections 7, 8, and 9, respectively. This is a very good and extensive review work, with 177 bibliographical references, which deserves to be published.

Dear reviewer,
We thank you for your feedback and valuable recommendations to improve our manuscript. Please, find below the answer to your different comments.
Some minor remarks:

1. The authors should review the text carefully to remove unnecessary hyphens and to correct mistyping.

The authors acknowledge the careful revision from the reviewer. All the minor typos that the reviewer mentioned were verified and corrected.

2. The list of abbreviations should be extended to include acronyms such as CPWR, ERMS, HHV,
SFC, TIMS, VGO, or VUV. In Abbreviations should be CCS instead of CSS.

The significance of theses acronyms was added and detailed in the abbreviation list.

superscripts. This was corrected and also in all the manuscript.

Reviewer 2 Report

The authors present a detailed review of the state of the art in HRMS analysis of bio-oils, highlighting many of the major analytical areas and the recent developments associated with those. Overall, this manuscript is promising and will be a benefit to the community pending revisions. 
The manuscript needs significant refinement prior to publication. Primarily, many of the references are cited in rather neutral terms - without highlighting the significance of the findings or contextualising them in the critical needs of (e.g.) industrial applications, instead just highlighting a key finding and trusting the reader knows how important that is. Further, there are numerous English language or typographic issues throughout, which should be addressed - some lead to ambiguous or incorrect interpretations of the text in their current state. 

Many of the concepts or comparisons within each section may be aided by a graphical representation - I am not sure that that included tables and graphics are the most efficient or beneficial in the context of the review. 

Specific comments:

·        Various grammatical omissions or errors, throughout. Manuscript needs a diligent proof-read. e.g.,

o   line 14 – ‘purposes’

o   line 15 – convention is ‘omics’ so it should be petroleomics, not petroleomic.

o Line 505, 507 use ‘petroleomics’ – so ensure the usage is standardized.

o   Line 16 ‘relies on’ or ‘relies upon’

·        Line 30 – What country? All countries? Or specific ones? Regardless, citation needed.

·        Table 1 needs precision standardized.

·        Line 131 – ‘sugary compounds’ is unusual phrasing. ‘Carbohydrate-like compounds’ would be more appropriate, and ‘great oxygen content’ could be defined.

·        Line 140 ‘et’ should be ‘and’ ?

·        Line 150 – ‘best’ is rather subjective

·        Line 159/160 – This sentence is ambiguously/confusingly phrased. 

·        Table 2 requires some citations or comment on source of information.

o   Further, there are mixed metrics which are hard or impossible to interpret – for example dynamic range is either ‘ND’ (acronym needs explanation), or a number, or an ADC value.  Either standardize them or remove them, otherwise its of very limited value.

o   Mass range should be in m/z and should show lower and upper bounds for all.

o   Vendor name isn’t shown, which may be helpful

o   The ‘scan rates’ (especially for Orbi and FTICR) are not necessarily true. Orbis can analyze at 40Hz, but they don’t get 1M resolution when they do. Equally, ICR at 1Hz is fairly arbitrary. If you want 20M at m/z 400 on a 7T 2xR you need hundreds of seconds per transient, i.e. 0.01Hz or less. But you can run an ICR at several Hz, too.

o   ‘timsTOF’ is stylized thus.

o   FTMS instruments don’t ‘scan’ – would be better to say ‘duty cycle’ or ‘measurement rate’

·        Line 189 – Is boron significant in this context? Is this a critical detection capability or just a curiosity?

·        Line 219 – ‘sugaric’ isn’t a word

·        Line 224 – typographic error? ‘… by. By comparison…’?

·        Section 2.3 – reference 61 (Qi and Volmer) also highlights the differences in ionization of lignin-degradation products when you use different matrices – this point seems to be omitted in lieu of ref 63 which is quoted that there aren’t ‘significant improvements’ compared to LDI results, but what does ‘significant improvements’ mean? Whats the objective measure?

·        (Secontion 2.4) Comparisons of LDI/MALDI ESI APPI APCI has also been done in the natural organic matter space

o   Anal. Chem. 2008, 80, 23, 8908–8919… Hertkorn et al  -

o   Analytica Chimica Acta Volume 866, 25 March 2015, Pages 48-58, Cao et al

o   Anal. Chem. 2017, 89, 8, 4382–4386 Blackburn and Kew

o   And for food products – e.g. Anal. Chem. 2018, 90, 19, 11265–11272, Kew at al.

o   These examples could also serve as useful reference points in the wider space as all contain lignin and carbohydrate derived compounds of diverse chemistries.

·        Line 312 – SARA? Should this be defined?

·        Table 3 – unclear what the significance of the nebulizing gas is? N2 is a conventional choice – was O2 necessary for that fraction? What happened if they didn’t use O2? Does O2 and APCI form ozone or nitrous oxides? Are these reacting with the analytes?

·        Section 5.1 – Ion mobility – lacking discussion on the limits of the resolution of this technique. It isn’t just about getting CCS values, but even being able to resolve the multitude of isomeric structures. Ensure that (in this section) ‘resolution’ or ‘resolving power’ are clarified if you mean ‘mass’ or ‘mobility’.

·        Table 4 formatting makes it tricky to read, could probably be reconsolidated and refocused. Perhaps two tables? One highlighting the different types of IM and their pros/cons, and one summarizing the applications of IM?

o   Mixed nomenclature/detail of MS types – ‘HRMS’ is quoted without information on the type of analyzer for ref 130, but ref 118 is quoted including the type and magnetic field strength (12T FTICR). Presumably all groups are using HRMS – but maybe worth discriminating between FTMS and QToF? Ref 126 quoted as ‘HDMS’ buts a qtof, ref 128 is just quoted as ‘MS’?

·        Line 495 – ‘increase’ should presumably be ‘increasing’? Significant difference in meaning in this context.

·        Line 497 – ‘activation’ is probably better than ‘excitation’

·        Line 503 – EID can be applied to singly charged ions. It’s not commonly done, but its possible.

·        Line 505 – important because they’re widely used, or because they’re the most insightful in the context of the analytical question? I would have thought (variable wavelength) UVPD would be super useful when studying such complex aromatic systems?

·        Section 6.1 – Carbohydrates. What types of carbohydrates are significant in biooils? Presumably cellulose (and thus glucose) is a significant feature in biooils? Are xylans particularly common? My understanding is that structural elucidation of isomeric sugars by MS/MS is very challenging – does that matter in this context? Or is confirming an analyte is a sugar enough?

·        Line 540 what is PO?

·        Section 6 may benefit from highlighting that isolating individual precursors in a complex mixture is very difficult – only in-cell isolation with FTICR is able to isolate an individual ion, but those measurements are very insensitive and slow and limit your MS/MS to in-cell methods. So, tandem MS on bio-oils may also be an informatics problem when you have to fragment multiple ions at once (in a DIA type approach).

·        Line 634, 639 – It should be “31P” not “13P”

·        Line 697 (and throughout) – by not using superscripts or subscripts, chemical formula can be tricky to read or ambiguous. This is critical to fix for publication.

·        Line 696-698 – may be helpful to explain what ‘near critical’ means.

·        Is the HDX 16/18O exchange chemically inert beyond the isotope exchange? Line 696-699 describe placing a complex mixture under high temperature and high pressure in aqueous environment, and I would have assumed some chemical reactions would occur beyond just isotope exchange?

·        Isotope exchange and labelling via derivatization has also been explored in organic matter, including with 2H, 13C, and 18O, which may be worth highlighting? E.g. the works of Zherebker, or the 2020 review by York and Bell highlights further examples of this. Environ. Sci. Technol. 2020, 54, 6, 3051–3063

o   Different samples, but similar chemistries and origins, so worth highlighting the precedent or complementary fields. For example, the iodomethane reaction you mention investigating in the future has been done on NOM in the literature already.

Reviewer 3 Report

The review by Vincent and co workers on the  Characterization of Complex Organic Mixtures in the Field of Sustainable Energy show various useful mass analytic techniques. Especially for the petrochemical and to characterize the higher molecular weight organic compounds this review will be helpful to find the exact technique. I strongly recommend accepting this review in Molecules so that it will have a great impact on finding the perfect analytical method on the characterization of the various complex molecules.

Before acceptance the author has to correct the few terminologies as below example.
Line 683 -Capture 2 (D2O) in subscripts. Line 697 - Capture the 18 labelled oxygen and other labels properly in superscripts/subscripts. Capture the exact molecular formulas like subscripts and superscripts.

Round 2

Reviewer 2 Report

The authors have addressed my comments and I think this is now ready for publication, pending very minor corrections, e.g.:

Reference 1 is not appropriately formatted ('1. Bp-Stats-Review-2021-Full-Report.Pdf. 906') 

Reference 14 may not be appropriately detailed. 
